# Bone Status in Patients with Phenylketonuria: A Systematic Review

**DOI:** 10.3390/nu12072154

**Published:** 2020-07-20

**Authors:** María José de Castro, Carmela de Lamas, Paula Sánchez-Pintos, Domingo González-Lamuño, María Luz Couce

**Affiliations:** 1Department of Pediatrics, University Clinical Hospital of Santiago de Compostela, 15706 Santiago de Compostela, Spain; mj.decastrol@gmail.com (M.J.d.C.); paula.sanchez.pintos@sergas.es (P.S.-P.); 2IDIS-Health Research Institute of Santiago de Compostela, 15706 Santiago de Compostela, Spain; 3CIBERER, Pabellón 11, 28029 Madrid, Spain; 4Faculty of Medicine, Santiago de Compostela University, 15782 Santiago de Compostela, Spain; carmeladelamas@gmail.com; 5Department of Pediatrics, University Hospital Marqués de Valdecilla, Instituto de Investigación Valdecilla (IDIVAL), University of Cantabria, 39005 Santander, Spain

**Keywords:** bone, bone mineral density, bone turnover markers, fractures, hyperphenylalaninemia, osteopenia

## Abstract

Phenylketonuria (PKU) is the most common inborn error of amino acid metabolism. Although dietary and, in some cases, pharmacological treatment has been successful in preventing intellectual disability in PKU patients who are treated early, suboptimal outcomes have been reported, including bone mineral disease. In this systematic review, we summarize the available evidence on bone health in PKU patients, including data on bone mineral density (BMD) and bone turnover marker data. Data from cohort and cross-sectional studies of children and adults (up to 40 years of age) were obtained by searching the MEDLINE and SCOPUS databases following Preferred Reporting Items for Systematic Reviews and Meta-Analyses (PRISMA) guidelines. For each selected study, quality assessment was performed applying the Risk Of Bias In Non-randomized Studies of Interventions (ROBINS I) tool. We found that mean BMD was lower in PKU patients than in reference groups, but was within the normal range in most patients when expressed as Z-score values. Furthermore, data revealed a trend towards an imbalance between bone formation and bone resorption, favoring bone removal. Data on serum levels of minerals and hormones involved in bone metabolism were very heterogeneous, and the analyses were inconclusive. Clinical trials that include the analysis of fracture rates, especially in older patients, are needed to gather more evidence on the clinical implications of lower BMD in PKU patients.

## 1. Introduction

Phenylketonuria (PKU, OMIM 261600) is an inborn error of phenylalanine (Phe) metabolism caused in most cases (98%) by an inherited deficiency in l-phenylalanine-4-hydroxylase (PAH; EC 1.14.16.1) activity, leading to elevated levels of Phe in body fluids [1]. Since the 1960s, worldwide implementation of newborn screening to detect PKU has enabled early diagnosis and treatment, preventing the gravest consequence of PKU: severe mental retardation [2,3]. PKU treatment mainly consists of lifelong restriction of Phe intake by limiting the amount of natural protein in the diet, combined with administration of low Phe amino-acid mixtures and special low protein foods [4,5]. Intake of glycomacropeptide or large neutral amino acids can also improve the variety and convenience of the dietary therapy [6]. Since 2007, a synthetic form of tetrahydrobiopterin (6R-BH4) has been used to treat selected patients who have moderate forms of PKU and respond to the BH4 loading test [7,8]. More recently, enzyme substitution therapy has been approved to treat PKU in patients aged 16 years or older [9]. The success of treatment has, however, led to the discovery of secondary issues in the life-long outcome of PKU patients, including nutritional deficiencies in minerals, vitamins, and long chain polyunsaturated fatty acids [10,11,12,13], behavioral impairment [13,14], and mineral bone disease [15,16]. Such micronutrient nutritional deficiencies will be highly dependent on the compliance to the treatment.

A long standing concern is that bone health in PKU patients is poorer than that of the general population [17], potentially leading to growth failure and fractures. However, studies conducted to date have produced conflicting findings in terms of bone mineral density (BMD), possibly due to differences in the age of the patients studied, the techniques to assess BMD, and the criteria applied [18]. In general, dual energy X-ray absorptiometry (DXA) of the lumbar spine and hip is the preferred method of evaluating BMD, as it provides the most reliable measurement for predicting fracture risk and monitoring treatment. DXA returns a T-score, which is the BMD value relative to that of a healthy 30-year old adult at peak BMD. There is consensus that spine and hip BMD measurements in postmenopausal women and men aged 50 years or older should be interpreted using the WHO T-score definitions of osteoporosis and osteopenia [19,20]. DXA also provides a patient Z-score, which reflects a value normalized to age and sex matched controls and adjusted for ethnicity or race. The International Society for Clinical Densitometry (ISCD) recommends use of DXA BMD Z-scores rather than T-scores in younger men, premenopausal women, and children, taking into account that diagnosis of osteoporosis in these groups should not be based on densitometric criteria alone [21] and should include the presence of a clinically significant fracture history.

It is increasingly acknowledged that the development of the osteoporotic state involves the interaction of multiple mechanisms. Understanding the pathogenesis of osteoporosis begins with understanding how bone formation and remodeling occur. For this reason, non-invasive biochemical markers have been developed and validated for the assessment of the bone formation (e.g., serum bone-specific alkaline phosphatase (BALP), osteocalcin (OC), and procollagen type I carboxyterminal propeptide (PICP)) and bone resorption (calcium:creatinine (Ca:Cr) ratio, collagen type I cross-linked c-telopeptide (CTX), pyridinoline cross-linked telopeptide domain (ICTP), and urinary n-telopeptide of type I collagen (NTX), hydroxyproline, and deoxypyridinoline (D-Pyr)) processes. These biomarkers can aid in risk assessment and serve as additional monitoring tools once treatment has been initiated [22,23,24].

The incidence, severity, and clinical implications of bone disease associated with PKU, as well as the underlying mechanisms remain to be fully elucidated. Proposed causes of bone disease in PKU include dietary deficiencies due to protein restriction, irregular diet compliance causing fluctuating blood Phe levels, sedentarism, and genetic factors linked to PKU that may impair bone health [25,26].

In this systematic review, we present a comprehensive overview of evidence from cohort and cross-sectional studies assessing bone status in PKU patients, including BMD, bone turnover markers, and serum levels of minerals and hormones implicated in bone metabolism.

## 2. Methods

Our systematic review sought to answer the following review question: “Is bone status impaired in patients with phenylketonuria?”. The review was carried out following PRISMA (Preferred Reporting Items for Systematic reviews and Meta-analyses) guidelines [27] and was registered in the International Prospective Register of Systematic Reviews (PROSPERO).

### 2.1. Literature Search

Once the review question was formulated, we proceeded to search PUBMED and SCOPUS databases using the following search terms: PUBMED, “Phenylketonurias” [Mesh] AND Bone”; SCOPUS, “Phenylketonurias AND Bone”. The bibliographies of the articles returned by each search were manually reviewed, as were other previously published reviews on the topic.

### 2.2. Inclusion and Exclusion Criteria

All observational studies comparing bone health status between PKU patients and healthy individuals, written in Spanish or English and published between 1 January 1900 and 31 March 2020, were considered for inclusion in this review. Observational studies that compared a healthy population with PKU patients whose intake of natural proteins was limited were also considered for inclusion. Studies excluded from our review were those in which PKU patients received either no treatment or non-dietary forms of treatment and those in which PKU patients had a concomitant disease that could interfere with the results.

### 2.3. Exposure

Dietary treatment, consisting of limiting natural protein intake, was the exposure studied. The purpose of the study was to assess the influence of such treatment on bone health in PKU patients. All studies that met the inclusion criteria and had a number of participants > 1 were included in the review process, regardless of follow-up duration or Phe levels.

### 2.4. Primary Outcome Measures

Four types of outcome measures were considered useful for the evaluation of bone status: bone mineral content (BMC) measurements; data on markers on bone formation and on resorption; and blood levels of minerals, hormones, and vitamins implicated in bone metabolism.

BMC data considered valid were those that included BMC or BMD measurements (absolute values or Z-scores) acquired by absorptiometry at the level of the whole body, lumbar spine, femoral neck, or extremities. Non measurable-data, such as radiographic evaluation of deformities, were excluded.

The following were considered valid markers of bone formation: serum alkaline phosphatase (ALP), BALP, OC, and PICP (expressed as Z-scores or absolute values). The following (expressed as Z-scores or absolute values) were taken as markers of bone resorption: serum CTX; osteoprotegerin (OPG); receptor activator of nuclear factor kappa-B ligand (RANKL); tartrate resistant acid phosphatase (TRAP) and ICTP; osteoclasts from peripheral blood mononuclear cells (PBMC); D-Pyr (in urine); Ca:Cr ratio; hydroxyproline:creatinine ratio; and pyridinoline:creatinine ratio.

For minerals, vitamins, and hormones associated with bone metabolism, absolute serum values of the following markers were considered valid: calcium, phosphorus, magnesium, vitamin D, and parathyroid hormone (PTH).

### 2.5. Study Selection

The 14 studies [28,29,30,31,32,33,34,35,36,37,38,39,40,41] finally included in our review were independently selected by two authors from the 299 articles identified during the bibliographic search. In cases where there was a lack of consensus, the remaining authors acted as arbitrators.

### 2.6. Data Extraction

Two authors independently (M.J.d.C. and C.d.L.) collected data from the selected articles. The following data were extracted from each study: number of participants by sex, number of participants with PKU, age, type of study, Phe levels, outcome measures, results, and conclusions. The remaining authors acted as arbitrators in cases in which there was a lack of consensus (less than 3% of conflicts observed).

### 2.7. Assessment of Risk of Bias

Risk of bias assessment was performed using the ROBINS-I (Risk Of Bias In Non-randomized Studies of Interventions) tool [42], which assesses the risk of the following types of biases: due to confounding; bias in the selection of participants for the study; bias in the classification of interventions; bias due to deviations from intended interventions; bias due to missing data; bias in outcome measurement; and bias in the selection of the reported results. For each study included in the review, the risk of each form of bias was classified as 1 of 5 possible levels: low, moderate, serious, critical, and uncertain (in the case of insufficient information).

## 3. Results

Figure 1 summarizes the process by which articles were selected for this systematic review. The SCOPUS search returned 108 articles, while the PUBMED search returned 190. One additional article, selected after manual review of the bibliography from other sources, was included. Of the 298 articles identified in database searches, ninety-five duplicate articles were excluded, and one-hundred sixty-seven were excluded due to a lack of relevance of the abstract (45 did not include bone health data; 40 due to inappropriate study characteristics; 39 were systematic or narrative reviews; 29 did not include a PKU patient population; 13 were preclinical studies; and 1 was published in a language other than English or Spanish). Of the 37 full-text articles reviewed, seventeen were excluded due to the lack of a control group; 4 due to unsuitable study characteristics; 1 due to the absence of bone health data; and 1 because PKU patients were not receiving any dietary treatment. Ultimately, fourteen [28,29,30,31,32,33,34,35,36,37,38,39,40,41] articles were selected for inclusion in this systematic review.

### 3.1. Study Characteristics

Table 1, Table 2, Table 3 and Table 4 summarize the main characteristics of the 14 [26,27,28,29,30,31,32,33,34,35,36,37,38,39] selected cohort and cross-sectional studies, which are ordered according to the age of the study population. Two cohort studies were included [32,39]. All studies [28,29,30,31,32,33,34,35,36,37,38,39,40,41] were published after 1992 and four [28,30,33,40] in the last 10 years. It should be highlighted that in the last decade, a significant improvement of the quality and diversity of protein substitutes together with a more universal recognition of the need to keep patients under follow-up with better metabolic control were verified. This is important because more recent studies can be measuring outcomes in patients with improved nutritional and clinical management practices. In total, the 14 studies included 327 PKU and 10 hyperphenylalaninemia (HPA) patients, of whom 54 participated in cross-sectional studies. The age of the study populations ranged from 0–40 years. All studies [28,29,30,31,32,33,34,35,36,37,38,39,40,41] involved a pediatric population, and eight also included adults [32,33,34,35,38,39,40,41]. Of the eight studies that measured bone mineral content [28,29,30,31,32,33,34,35], 6 employed DXA [28,29,30,31,33,34], 1 single photon absorptiometry [32], and 1 quantitative computed tomography [35]. Ten studies evaluated bone formation markers [26,28,29,30,33,34,35,36,37,38], all of which were measured in serum (4 ALP [28,30,32,37], 4 BALP [31,36,38,40], 7 OC [30,31,35,36,38,39,40], 1 PICP [30], and 1 C-terminal propeptide [35]). Of the nine studies that evaluated bone resorption markers, five included serum markers [30,31,36,40,41] (1 measured CTX [36], 2 OPG [30,40], 1 RANKL [30], 1 TRAP [31], 1 osteoclasts from PBMC cultures [41], and 1 ICTP [40]), and six included urinary markers [30,31,35,37,38] (4 Ca:Cr ratio [31,37,38,40], 3 D-Pyr [30,38,40], 1 hydroxyproline:creatinine ratio [35], 1 pyridinoline:Cr ratio [35], and 1 NTx [40]). Six studies measured serum mineral and hormone levels [28,31,32,37,39,40]; 5 measured Ca [29,31,32,37,39] and P [28,31,32,37,39], 4 Mg [31,32,37,39], 3 PTH and 25-OH vitamin D [31,37,40], and 4 25-OH vitamin D [31,32,37,40].

### 3.2. Phenylketonuria and Bone Mineral Content

Eight [28,29,30,31,32,33,34,35] articles included in this systematic review assessed the effects of PKU on bone mineral content. Seven of the studies were cross-sectional [28,29,30,31,33,34,35], and one [32] was a cohort study. Six articles [28,29,30,31,33,34] found significant differences in BMC (total BMC [34]) or BMC measured in the lumbar region [28,29,30,31,33], femoral neck [30,33], or upper extremity, lower extremity, 1/3 radius and 1/10 radius [31]. All studies observed significant differences in BMC between PKU patients and the healthy population, as evidenced by a lower BMC in PKU patients. Five of the studies employed DXA [28,29,30,31,33], and one used quantitative computed densitometry [34]. Three of the studies that used DXA reported ZZ-scores [30,33,35]: although all reported significantly lower BMC values in PKU patients versus the healthy population, unexpectedly, PKU patients did not show a higher frequency of bone mass below the expected range for age (defined as Z-score values of −2.0 SD or lower). The only study that used DXA, but did not find significant differences in BMC between groups [35] included seven patients with PKU and 10 with HPA, which is associated with milder alterations in Phe levels and requires less severe dietary restriction.

### 3.3. Phenylketonuria and Bone Turnover Markers

Eleven [28,30,31,32,35,36,37,38,39,40,41] studies included in this review evaluated bone turnover markers. Ten measured bone formation markers [28,30,31,32,35,36,37,38,39,40] and eight bone resorption markers [30,31,35,36,37,38,40,41]. Two of the 11 studies [32,39] were cohort studies. Of the studies that measured bone formation markers, eight reported significant differences between PKU patients and healthy controls [28,30,31,32,36,37,38,39]. In seven of the eight studies, bone formation markers were lower in PKU patients than healthy controls [30,31,32,36,37,38,39], while the remaining study, which included the youngest patients (3–4 years), reported significantly higher levels of serum ALP in PKU patients [28]. Of the studies that measured bone resorption markers, six reported significant differences between groups [30,36,37,38,40,41]. In five of the six studies, bone resorption markers were increased in PKU patients versus healthy controls [30,37,38,40,41], while the remaining study, which included the youngest patients (4.5 years of age), reported significantly lower levels of serum CTX and OPG in PKU patients [36]. It should be noted that the only study that included both PKU and HPA patients [35] found no significant difference in either bone formation or bone resorption markers between PKU patients and healthy controls.

### 3.4. Phenylketonuria and Serum Minerals and Hormones

Six [28,31,32,37,39,40] studies included in this review measured serum levels of minerals and hormones involved in bone metabolism. All studies included PKU patients, and none included HPA patients. Two [32,39] were cohort studies. While all found significant differences between PKU patients and healthy controls, the results were very heterogeneous. In PKU patients, two studies reported higher Ca levels [28,37] and one higher Mg levels [37]. One study reported lower Ca levels [31], 2 lower Mg levels [31,32], 1 lower P levels [39], and 1 lower vitamin D levels [32] in PKU patients versus healthy controls. One study [40] observed sex-related differences in PTH and 25-OH vitamin D levels, with higher levels in females and lower levels in males.

### 3.5. Risk of Bias Assessment

For all articles included in the review, the risk of bias due to deviations from intended interventions and due to measurement of outcomes was considered low (Appendix A). The risk of bias due to missing data was low for 50% of the studies and uncertain for the remaining 50%. The risk of bias in the selection of participants and of bias in classification of the intervention was low in 43% of studies and moderate in the remaining 57% owing to the joint analysis of cases with a large difference in evolution time and exposure to a low protein diet. The risk of bias due to confounding was moderate for all studies included in the review, due to the presence of different confounding factors such as the possible impact of the disease itself and of its treatment on bone health status. The risk of bias in the selection of the reported results was deemed uncertain for all studies included in this systematic review. The studies by Wang [28], Ambroszkiewicz [36], and Pérez-Dueñas [39] were those for which the risk of biased results was lowest. In these studies, the risk of all types of bias studied was low, except for the risk of bias due to confounding (which was moderate) and the risk of bias in the selection of the reported results (which was uncertain). Four of the studies [29,30,31,40] that were ultimately included in the systematic review had a greater risk of bias than the others. The analysis of these articles revealed that, of the seven types of bias studied, there was a low risk for only two types of bias (bias due to deviations from intended interventions and bias in measurement of outcomes); an uncertain risk of two forms (bias due to missing data and bias in the selection of the reported results); and a moderate risk of three remaining forms of bias analyzed. Further information on the risk-of-bias assessment can be found in the Appendix A.

## 4. Discussion

This systematic review of cross-sectional and cohort studies assesses bone mineral status, levels of bone turnover markers, and levels of minerals and elements related to bone health in PKU patients. The results suggest that mean BMD is lower in PKU patients compared with reference groups, despite being within the normal range in most patients. Moreover, we observed a trend towards an imbalance between bone formation and bone resorption, suggesting that bone turnover may be skewed in favor of bone removal. Analyses of data on serum levels of minerals and hormones involved in bone metabolism were inconclusive.

PKU patients are detected early by newborn screening, put in place in the 1960s, and are treated predominantly using a Phe-restricted diet, which limits the intake of natural protein. As the first early treated PKU patients age, concerns have arisen about the long-term consequences of PKU and its treatment [10,11,12,13,14], including deleterious effects on bone health [15,16]. Osteoporosis is a silent disease until it is complicated by fractures that can occur due to minimal or no trauma. It can be prevented, diagnosed, and treated before complications occur, and therefore, early detection in vulnerable populations should be mandatory [43,44]. In this review, six of the eight studies that evaluated BMD used DXA technology to analyze the lumbar spine and neck of the femur, and none included menopausal women or adult men aged over 50 years. Those studies reported reduced bone mineralization in PKU patients versus the general population. However, only three of the six studies [30,33,35] reported BMD Z-scores, which were within the normal range. This observation casts some doubt on the clinical relevance of these findings, particularly given that none of the selected studies compared fracture history, which is essential to qualify for the diagnosis of osteopenia and osteoporosis together with Z-scores. McMurry et al., Schwan et al., and Fernandez et al. [32,34,35] compared BMD in PKU patients from infancy through adolescence to early adulthood, and all reported that mineralization defects began in childhood and became more pronounced in adolescents and older subjects, suggesting either a cumulative disease- or diet-related effect over time or deterioration of dietary control in this age group. In line with this hypothesis, a study by Greeve et al. [45] comparing the history of fractures in a group of PKU patients aged 0.3–33.6 years and their healthy siblings reported a significantly higher risk of fracture in PKU patients over eight years of age.

The process of bone remodeling continually removes older bone and replaces it with new bone, thereby maintaining a healthy skeleton. Bone loss occurs when this balance becomes skewed, resulting in greater bone resorption than formation. Currently available biochemical markers for the assessment of bone turnover include enzymes and nonenzymatic peptides derived from the cellular and noncellular compartments of bone and are used to assess fracture risk and monitor treatment response in clinical settings. The studies included in this review revealed a trend towards increased levels of bone resorption markers and decreased levels of bone formation markers in patients with PKU [30,31,32,36,37,38,39,40,41]. Interestingly, only two studies compared both types of bone turnaround markers from pediatric age to early adulthood, including adolescence [35,38]. Millet et al. [38] reported significantly higher levels of bone resorption markers in PKU patients, independent of age, and lower levels of bone formation markers only in the oldest group. These results suggest that bone formation is active in childhood, but deteriorates in adult PKU patients, while resorption appears to remain a constant process throughout the disease course. Fernandez et al. [35] also reported similar findings throughout the life of patients with disorders of phenylalanine metabolism and found that compared with patients with benign hyperphenylalaninemia, PKU patients had the highest risk of developing osteoporosis/osteopenia due to a decrease in overall bone turnover. Success of current efforts to harmonize markers of bone turnover could lead to the use of markers to predict fracture risk independently of BMD based on micro-architectural alterations affecting bone quality [19].

Data from the studies included in this review on serum levels of minerals and hormones such as calcium, magnesium, phosphorus, vitamin D, and PTH were not conclusive, owing to the high variability between studies. It should be noted that low Phe formulas are fortified with minerals and vitamins including calcium and vitamin D, and PKU patients who are more compliant with Phe-restricted diets may have normal levels of calcium and vitamin D, despite a low BMD. Several studies have shown that adequate intake of Ca, P, and vitamin D is not sufficient for normal bone development in individuals with a decreased intake of natural protein [44,46], which plays a more important role in BMD development in PKU patients [13].

Whether these observations are caused by a Phe-restricted diet or reflect a toxic effect of the disease itself remains unclear; in this review, it was not possible to determine the precise intake of either natural protein or medical food or Phe levels in PKU patients. However, both PKU and a Phe-restricted diet may lead to a more pronounced decrease in BMD in older patients and, therefore, to increased incidence of osteoporosis in adult PKU patients. It should be underlined that the availability of protein substitutes and special low protein foods in different countries will determine different levels of diet compliance and adherence, which will influence micronutrient status and bone status.

Our findings underscore the need for future studies to homogeneously report the number of PKU subjects with spine, femoral neck, and total hip Z-scores < −2.0 SD or T-scores below −1.0 SD, in order to define more precisely the presence of osteopenia and osteoporosis. Moreover, the interaction of amino acids with hydroxyapatites of different calcium content could be of some clinical relevance. Thus, to clarify the clinical implications of the observed mineralization defects in PKU patients, the rate of fractures should be analyzed and matched with healthy controls, especially in older patients given the apparent age-associated increase in mineralization defects.

## 5. Conclusions

The findings of this systematic review support the view that BMD is reduced in PKU patients compared with the healthy population, although this does not translate to an increased prevalence of bone mass below the expected range for age as defined by Z-scores of −2.0 SD or lower. Furthermore, we observed a trend towards an imbalance between bone formation and bone resorption, favoring bone removal. Analysis of data on serum levels of minerals and hormones involved in bone metabolism was inconclusive.

## Figures and Tables

**Figure 1 nutrients-12-02154-f001:**
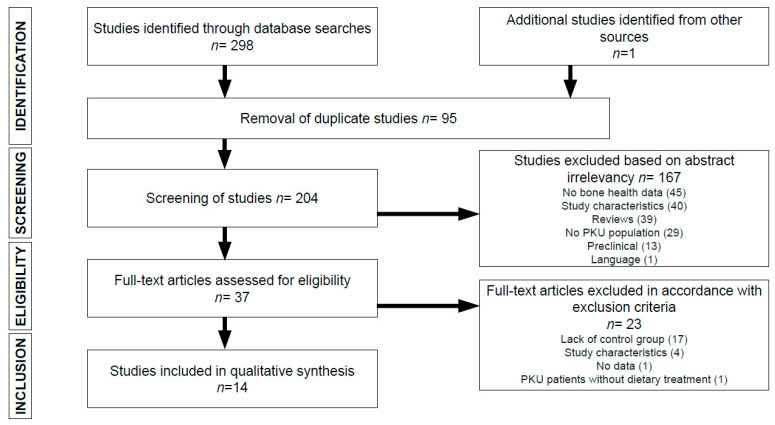
Flowchart depicting literature search process. PKU, phenylketonuria.

**Table 1 nutrients-12-02154-t001:** Bone mineral content in patients with phenylketonuria.

Reference Country	*n*	Age ^1^	Trial Type/ Phe Levels ^2^	Outcome Measure	Results ^3^	Conclusions
Wang et al., China (2017) [28]	105 41 PKU	3–4	Cross-sectional/ PG: 43–1776 μmol/L	Mean lumbar L1–L4 BMD (DXA) in g/cm^2^	L1–L4 BMD: (3 years) PG 0.49 ± 0.06; CG 0.54 ± 0.05 (*p* = 0.03); (4 years) PG 0.51 ± 0.09; CG 0.57 ± 0.06 (*p* = 0.01)	Significantly lower lumbar BMD
Allen et al., Australia (1994) [29]	127(50 F) 32 PKU	PG 7.7 ± 2.3 CG 8.1 ± 2.1	Cross-sectional	Mean total body and lumbar L2–L4 BMD (DXA) in g/cm^2^	TBMD: PG 0.77 ± 0.08; CG 0.81 ± 0.07 (*p* < 0.05) L2-L4 BMD: PG 0.61 ± 0.10; CG 0.78 ± 0.06 (*p* < 0.05)	Significantly lower lumbar and total BMD
Koura et al., Egypt (2014) [30]	77 (34 F) 33 PKU	PG 8.4 ± 4.6 CG 8.5 ± 3.3	Cross-sectional	Mean total body BMC and Z-score BMC, femoral neck and lumbar spine BMD, and Z-score BMD (DXA) in g and g/m^2^	TBMC: PG 1072 ± 596; CG 1269 ± 557 Z-TBMC: PG −0.1 ± 1.2; CG 1.1 ± 1 (*p* < 0.001) Femoral neck BMD: PG 0.5 ± 0.1; CG 0.7 ± 0.1 (*p* < 0.001) Femoral neck Z-BMD: PG −0.6 ± 0.7; CG 0.02 ± 0.4 (*p* < 0.001) Lumbar BMD: PG 0.5 ± 0.1; CG 0.5 ± 0.1 Lumbar Z-BMD: PG −0.3 ± 0.6; CG 0.06 ± 0.6 (*p* = 0.01)	Significantly lower Z-TBMC, femoral neck BMD and z-BMD, and lumbar z-BMD
Hillman et al., USA (1996) [31]	22 11 PKU	PG 10.9 ± 4.2 CG 11.4 ± 4.2	Cross-sectional PG: 9.9 ± 9.5 mg/dL	Mean total body, lumbar spine, upper extremity, lower extremity, 1/3 radius and 1/10 radius BMD (DXA) in g/cm^2^	TBMD: PG 0.80 ± 0.10; CG 0.88 ± 0.18 Lumbar BMD: PG 0.61 ± 0.15; CG 0.72 ± 0.24 (*p* = 0.04) Upper BMD: PG 1.25 ± 0.16; CG 1.37 ± 0.28 Lower BMD: PG 1.56 ± 0.3; CG 1.87 ± 0.56 (*p* = 0.03) 1/3 radius BMD: PG 0.48 ± 0.08; CG 0.53 ± 0.12 1/10 radius BMD: PG 0.32 ± 0.07; CG 0.36 ± 0.11	Significantly lower lumbar BMD and lower extremities BMD.
McMurry et al., USA (1992) [32]	190 (91 F) 26 PKU	PG 1.9–25.5 CG 3–16	Cohort study PG (1–5 years): 581 ± 121 PG (6–11 years): 1041 ± 188 PG (>11 years): 1629 ± 170 μmol/L	Mean total body BMC and BMD (non-dominant radius single photon absorptiometry) in g/cm and g/cm^2^	BMC: (1–5 years) PG 0.26 ± 0.02; CG 0.26 ± 0.07; (6–11 years) PG 0.42 ± 0.01; CG 0.45 ± 0.1; (>11 years) PG 0.70 ± 0.05; CG 0.72 ± 0.15 BMD: (1–5 years) PG 0.28 ± 0.01; CG 0.29 ± 0.06; (6–11 years) PG 0.4 ± 0.007; CG 0.41 ± 0.06; (>11 years) PG 0.57 ± 0.03; CG 0.58 ± 0.008	No significant differences
Koura et al., Egypt (2011) [33]	74 32 PKU	PG 3–19	Cross-sectional	Mean femoral neck and lumbar L2–4 BMC, BMD, and Z-score BMD (DXA) in g and g/cm^2^	Femoral neck BMC: PG 2.0 ± 0.18; CG 0.7 ± 0.07 Femoral neck BMD: PG 0.6 ± 0.03; CG 0.7 ± 0.02 (*p* < 0.0001) Femoral neck Z-BMD: PG −0.7 ± 0.12; CG 0.03 ± 0.08 (*p* < 0.0001) Lumbar L2–4 BMC: PG 14.9 ± 1.65; CG 17.4 ± 1.29 Lumbar L2–4 BMD: PG 0.5 ± 0.03; CG 0.6 ± 0.02 Lumbar L2–4 Z-BMD: PG −0.4 ± 0.12; CG 0.1 ± 0.11 (*p* = 0.01)	Significantly lower femoral neck BMD, femoral neck Z-BMD, and lumbar L2–4 Z-BMD
Schwahn et al., (1998) [34]	28 (12 F) 14 PKU	5–28	Cross-sectional	Mean TBMD and SBMD (pQCT) in mg/cm^3^	TBMD: PG 290.9 ± 64.4; CG 305.4 ± 67.6 SBMD: PG 139.7 ± 23.5; CG 169.3 ± 31.5 (*p* < 0.01)	Significantly lower SBMD
Fernández et al., Germany (2005) [35]	92 7 PKU 10 HPA	PKU 6–29 HPA 4–16 CG 0.5–14	Cross-sectional	Mean Z-score TBMD (DXA)	Z-TBMD: PKU −0.45 ± 0.83; HPA 0.45 ± 0.86	No significant differences

BMC, bone mineral content; BMD, bone mineral density; CG, control group; DXA, dual energy X-ray absorptiometry; F, female; HPA, hyperphenylalaninemia; n.s.: not significant, PG, patient group; Phe, phenylalanine; PKU, phenylketonuria; pQCT, quantitative computed densitometry of non-dominant radius; SBMD, spongy bone mineral density; SPA, single photon absorptiometry; TBMD, total bone mineral density; y, years. ^1^ Values represent the range or mean ± SD in years, as reported in the corresponding article. ^2^ Blood phenylalanine levels represent range as reported in the corresponding articles. ^3^ Values represent mean ± SD as reported in the corresponding article.

**Table 2 nutrients-12-02154-t002:** Bone formation in patients with phenylketonuria.

Reference Country	*n*	Age ^1^	Trial Type Phe Levels ^2^	Outcome Measure	Results ^3^	Conclusions
Wang et al., China (2017) [28]	105 41 PKU	3–4 years	Cross-sectional PG: 43-1776 μmol/L	Mean serum ALP (ELISA) in IU/L	ALP: (3 years) PG 209 ± 54; CG 134 ± 42 (*p* < 0.01); (4 years) PG 203 ± 51; CG 138 ± 51 (*p* = 0.01)	Significantly higher serum ALP
Ambroszkiewicz et al., Poland (2004) [36]	64 (44 F) 37 PKU	PGG 4.5 PGB 6.0 CG 5.9	Cross-sectional PGG: 189 ± 64 PGB: 649 ± 140 μmol/L	Mean serum OC (ELISA) in µg/L and BALP (RIA) in IU/L	OC: PGG 67.1 (42–140); PGB 80 (43–148); CG 102.8 (79–121) (*p* < 0.05) BALP: PGG 93.8 (75–141); PGB 102.5 (74–145); CG 110.2 (89–129)	Significantly lower serum OC
Koura et al., Egypt (2014) [30]	77 (34 F) 33 PKU	PG 8.4 ± 4.6 CG 8.5 ± 3.3	Cross-sectional	Mean serum OC (ELISA) in mg/dL, ALP in IU/L, and PICP (ELISA) in ng/mL	OC: PG 13.9 ± 12.9; CG 43.4 ± 34.5 (*p* < 0.001) ALP: PG 121.6 ± 46; CG 152 ± 43.1 (*p* = 0.005) PICP: PG 283.4 ± 114.7; CG 270.7 ± 89.6	Significantly lower serum OC and ALP
Al-Qadreh, Greece (1998) [37]	98 (56 F) 48 PKU	PG 8.8 ± 3.7 CG 9 ± 3.5	Cross-sectional PG: 11.1 ± 6.6 mg/dL	Mean serum ALP in IU/L	ALP: PG 73.3 ± 4.9; CG 89 ± 3.6 (*p* = 0.01)	Significantly lower serum ALP
Hillman et al., USA (1996) [31]	22 11 PKU	PG 10.9 ± 4.2 CG 11.4 ± 4.2	Cross-sectional PG: 9.9 ± 9.5 mg/dL	Mean serum OC (RIA) in µg/L, BALP (colorimetry) in IU/L, and PICP (RIA)	OC: PG 6.1 ± 6.3; CG 13.1 ± 2.0 (*p* < 0.01) BALP: PG 72 ± 30; CG 126 ± 43 (*p* < 0.001) PICP: PG 290 ± 174; CG 400 ± 159	Significantly lower serum OC and BALP
Millet et al., Spain (2005) [38]	226 (120 F) 46 PKU	PG 17.5 (4–38) CG 8.99 (0–26)	Cross-sectional	Mean serum OC (chemiluminescent assay), and BALP (IRMA) in µg/L	OC: (6–8 years) PG 27.5 (3.6–50); CG 24.1 (3.4–84); (9–15 years) PG 29.9 (9–70); CG 42.6 (5.5–77); (>15 years) PG 7.2 (2.7–35); CG 11.4 (2.7–14) BALP: (6–8 years) PG 64.3 (31.9–89); CG 49.9 (21–114); (9–13 years) PG 69.2 (36–99); CG 51.9 (23–79) (*p* = 0.016); (14–18 years) PG 30.9 (13–48); CG 27.7 (14–51); (>18 years) PG 14.2 (8.8–39); CG 18.8 (9–21) (*p* = 0.003)	Significantly higher serum BALP in 9–13 years PKU patients and significantly lower serum BALP in >18 years PKU patients and OC in >15 years patients
McMurry et al., USA (1992) [32]	190 (91 F) 26 PKU	PG 1.9–25.5 CG 3–16	Cohort study PG (1–5 years): 581 ± 121 PG (6–11 years): 1041 ± 188 PG (>11 years): 1629 ± 170 μmol/L	Mean serum ALP in µkat/L	ALP: (1–5 years) PG 1.2 ± 0.1; CG 2.1 ± 0.2 (*p* < 0.02); (6–11 years) PG 0.9 ± 0.1; CG 1.6 ± 0.1 (*p* < 0.001); (>11 years) PG 0.6 ± 0.1; CG 1.5 ± 0.1 (*p* < 0.001)	Significantly lower serum ALP
Pérez-Dueñas et al., Spain (2002) [39]	97 28 PKU	PG 18 (10–33) CG 10–34	Cohort study	Mean serum OC (chemiluminescent assay) and BALP (IRMA) in µg/L	OC: (11–5 years) PG 47.5(19–73); CG 48(15–78); (19–33 years) PG 12.9 (9.5–18.9); CG 9.9 (4.4–26) BALP: (11–15 years) PG 50.8 (22.6–76); CG 42 (15–84); (19–33 years) PG 11.4 (8.8–13); CG 18.9 (16–22) (*p* < 0.0001)	Significantly lower serum BALP in 19–33 years patient group
Fernández et al., Spain (2005) [35]	92 7 PKU 10 HPA	PKU 6–29 HPA 4–16 CG 0.5–14	Cross-sectional	Mean Z-score serum OC and C-terminal propeptide (enzyme immunoassay)	Z-OC: PKU 0.75 ± 1.26; HPA 0.67 ± 0.92 Z-C-term: PKU −0.23 ± 0.49; HPA 0.91 ± 1.16	No significant differences
Nagasaka et al., Japan (2011) [40]	70 (43 F) 34 PKU	PG 20–35 CG 19–40	Cross-sectional	Mean serum OC (RIA) in ng/mL and BALP (ELISA) in IU/L	OC: FPG 5.6 ± 0.7; FCG 5.9 ± 0.5; MPG 5.4 ± 1.0; MCG 5.5 ± 0.6 BALP: FPG 22.7 ± 2.2; FCG 21.7 ± 2.5; MPG 28.5 ± 2.7; MCG 25.4 ± 2.7	No significant differences

ALP, alkaline phosphatase; BALP, bone alkaline phosphatase; CG, control group; ELISA, enzyme-linked immunosorbent assay; F, female; FCG, female control group; FPG, female patient group; HPA, hyperphenylalaninemia; IRMA, immunoradiometric assay; MPG, male patient group; OC, osteocalcin; PG, patient group; PGB, bad control patient group (increased serum phenylalanine); PGG, good control patient group (normal serum phenylalanine); PICP, procollagen type I carboxyterminal propeptide; PKU, phenylketonuria; RIA, radioimmunoassay; y, years. ^1^ Values represent the range, mean (range), or the mean ± SD in years, as reported in the corresponding article. ^2^ Blood phenylalanine levels represent range as reported in the corresponding articles. ^3^ Values represent the mean ± SD or mean (range) as reported in the corresponding article.

**Table 3 nutrients-12-02154-t003:** Bone resorption in patients with phenylketonuria.

Reference Country	*n*	Age ^1^	Trial Type Phe Levels ^2^	Outcome Measure	Results ^3^	Conclusions
Ambroszkiewicz et al., Poland (2004) [36]	64 (44 F) 37 PKU	PGG 4.5 PGB 6.0 CG 5.9	Cross-sectional PGG: 189 ± 64 PGB: 649 ± 140 μmol/L	Mean serum CTX (ELISA) in mg/L and OPG (RIA) in pmol/L	CTX: PGG 1322 (1017–2871); PGB 1685 (1096–2762); CG 2030 (1363–2815) (*p* < 0.01) OPG: PGG 3.58 (2.32–4.59); PGB 3.33 (2.37–5.01); CG: 4.46 (2.34–5.64) (*p* < 0.01)	Significantly lower serum CTX and OPG
Koura et al., Egypt (2014) [30]	77 (34 F) 33 PKU	PG 8.4 ± 4.6 CG 8.5 ± 3.3	Cross-sectional	Mean serum OPG and RANKL (ELISA) in ng/mL and urinary D-Pyr (ELISA) in mmol/creatinine mmol	OPG: PG 4 ± 0.8; CG 3.3 ± 2.3 RANKL: PG 1.0 ± 0.2; CG 0.1 ± 0.07 (<0.001) D-Pyr: PG 32.3 ± 15; CG 68.1 ± 30.7 (<0.001)	Significantly higher serum RANKL and significantly lower urinary D-Pyr
Al-Qadreh Greece (1998) [37]	98 (56 F) 48 PKU	PG 8.8 ± 3.7 CG 9 ± 3.5	Cross- sectional PG: 11.1 ± 6.6 mg/dL	Mean urinary Ca:Cr ratio	UCa:Cr: PG 0.46 ± 0.05; CG 0.22 ± 0.01 (<0.001)	Significantly higher urinary Ca:Cr ratio
Hillman et al., USA (1996) [31]	22 11 PKU	PG 10.9 ± 4.2 CG 11.4 ± 4.2	Cross-sectional PG: 9.9 ± 9.5 mg/dL	Mean serum TRAP (enzymatically) in IU/L and urinary Ca:Cr ratio	TRAP: PG 11.4 ± 3.3; CG 12.0 ± 5.0 UCa:Cr: PG 0.17 ± 0.22; CG 0.12 ± 0.09	No significant differences
Porta et al., Italy (2008) [41]	40 20 PKU	14 ± 7.1	Cross-sectional	Mean number of osteoclasts from PBMC cultures	Osteoclasts: PG 159.9 ± 79.5; CG 87.8 ± 44.7 (*p* = 0.001)	Significantly higher spontaneous osteoclastogenesis from PBMCs
Millet et al., Spain (2005) [38]	226 (120 F) 46 PKU	PG 17.5 (4–38) CG 8.99 (0–26)	Cross-sectional	Mean urinary D-Pyr (chemiluminescent assay) in μmol/mol creatinine and Ca:Cr ratio	D-Pyr: (4–6 years) PG 33.6 (29.1–55.0); CG 27.6 (12.4–45.1); (7–11 years) PG 29.6 (18–40); CG 22 (10.9–36.7) (*p* = 0.005); (12–14 years) PG 30.7 (17.3–34.4); CG 14.3 (6.9–30) (*p* = 0.004); (15–17 years) PG 11.4 (6.5–7.2); CG 8.1 (3.6–20.5); (>18 years) PG 6.7 (3.6–13.4); CG 5.4 (3.6–10.9) (*p* = 0.031) UCa:Cr: (4–6 years) PG 0.38 (0.36–1.8); CG 0.29 (0.01–0.94); (7–11 years) PG 0.43 (0.03–1.03); CG 0.31 (0.03–0.75); (12–14 years) PG 0.17 (0.03–0.56); CG 0.24 (0.02–0.84); (15–17 years) PG 0.33 (0.05–0.57); CG 0.24 (0.06–0.84); (>18 years) PG 0.49 (0.25–0.81); CG 0.33 (0.05–0.56) (*p* < 0.001).	Significantly higher D-Pyr in 7–14 years and >18 years PKU patients and significantly higher urinary Ca:Cr ratio in >18 years PKU patients
Fernández et al., Spain (2005) [35]	92 7 PKU 10 HPA	PKU 6–29 HPA 4–16 CG 0.5–14	Cross-sectional	Mean Z-score urinary hydroxyproline:creatinine ratio (HPLC) and pyridinoline:creatinine ratio (enzyme immunoassay)	z-H/Cr: PKU −1.07 ± 0.98; HPA −0.43 ± 1.18 z-P/Cr: PKU −0.21 ± 1.26; HPA 1.07 ± 1.25	No significant differences
Nagasaka et al., Japan (2011) [40]	70 (43 F) 34 PKU	PG 20–35 CG 19–40	Cross-sectional	Mean serum ICTP (RIA) in ng/mL, OPG (ELISA) in pmol/L, urinary D-Pyr (ELISA) and NTx (ELISA) in nmol/mmol, and urinary Ca:Cr ratio	ICTP: FPG 4.6 ± 0.2; FCG 3.0 ± 0.2 (*p* <0.001); MPG 4.3 ± 0.3; MCG 3.0 ± 0.2 (*p* <0.01) OPG: FPG 3.3 ± 0.3; FCG 4.7 ± 0.4 (*p* <0.001); MPG 3.1 ± 0.2; MCG 4.3 ± 0.2 (*p* <0.01) D-Pyr: FPG 7.3 ± 0.5; FCG 4.9 ± 0.4 (*p* <0.001); MPG 5.2 ± 0.5; MCG 3.8 ± 0.6 (*p* <0.01) NTx: FPG 47.8 ± 6.1; FCG 31.7 ± 5.1 (*p* <0.001); MPG 54.7 ± 12.1; MCG 38.3 ± 10.5 (*p* <0.01) UCa:Cr: FPG 0.4 6 ± 0.08; FCG 0.33 ± 0.88 (*p* <0.001); MPG 0.42 ± 0.1; MCG 0.3 ± 0.07 (*p* <0.05)	Significantly higher ICTP, D-Pyr, NTx, and urinary Ca:Cr ratio; significantly lower serum OPG

Ca, calcium; CG, control group; Cr, creatinine; CTX, collagen type I cross-linked c-telopeptide; D-Pyr, urinary deoxypyridinoline; ELISA, enzyme-linked immunosorbent assay; F, female; FPG, female patient group; HPA, hyperphenylalaninemia; HPLC, high performance liquid chromatography; ICTP, pyridinoline cross-linked telopeptide domain; MPG, male patient group; NTx, urinary n-telopeptide of type I collagen; OPG, osteoprotegerin; PBMC, peripheral blood mononuclear cells; PG, patient group; PGB, bad control patient group (increased serum phenylalanine); PGG, good control patient group (normal serum phenylalanine); PKU, phenylketonuria; RANKL, receptor activator of nuclear factor kappa-B ligand; TRAP, tartrate resistant acid phosphatase; UCa:Cr, urinary Ca:Cr ratio; y, years. ^1^ Values represent the range, the mean (range), or the mean ± SD in years, as reported in the corresponding article. ^2^ Blood phenylalanine levels represent range as reported in the corresponding articles. ^3^ Values represent the mean ± SD or mean (range) as reported in the corresponding article.

**Table 4 nutrients-12-02154-t004:** Serum minerals and hormones in patients with phenylketonuria.

Reference Country	*n*	Age ^1^	Trial Type Phe Levels ^2^	Outcome Measure	Results ^3^	Conclusions
Wang et al., China (2017) [28]	105 41 PKU	3–4	Cross-sectional PG: 43-1776 μmol/L	Mean serum Ca and P (colorimetry) in mmol/L	Ca: (3 years) PG 2.42 ± 0.09; CG 2.37 ± 0.12; (4 years) PG 2.41 ± 0.13; CG 2.32 ± 0.12 (*p* = 0.029) P: (3 years) PG 1.65 ± 0.31; CG 1.53 ± 0.22; (4 years) PG 1.49 ± 0.21; CG 1.57 ± 0.22	Significantly higher serum Ca in PKU patients
Al-Qadreh Greece (1998) [37]	98 (56 F) 48 PKU	PG 8.8 ± 3.7 CG 9 ± 3.5	Cross sectional PG: 11.1 ± 6.6 mg/dL	Mean serum Ca, Mg, and P in mmol/L, PTH in pmol/L, and 25-OHD in nmol/L	Ca: PG 2.51 ± 0.02; CG 2.47 ± 0.02 (*p* = 0.04) Mg: PG 0.94 ± 0.01; CG 0.86 ± 0.01 (*p* < 0.001) P: PG 1.63 ± 0.03; CG 1.65 ± 0.04 PTH: PG 16.6 ± 2.6; CG 23.0 ± 2.4 25-OHD: PG 45.3 ± 3.8; CG 49.16 ± 2.54	Significantly higher serum Ca and Mg
Hillman et al., USA (1996) [31]	22 11 PKU	PG 10.9 ± 4.2 CG 11.4 ± 4.2	Cross-sectional PG: 9.9 ± 9.5 mg/dL	Mean serum Ca, Mg (flame atomic absorption) and P (calorimetry) in mg/dL, PTH (RIA), and 25-OHD (immunoassay)	Ca: PG 9.1 ± 0.9; CG 10.4 ± 1.9 (*p* < 0.01) Mg: PG 1.67 ± 0.14; CG 2.07 ± 0.16 (*p* < 0.001) P: PG 5.6 ± 7.1; CG 5.5 ± 0.9 PTH: PG 24.1 ± 9.1; CG 25.0 ± 9.3 25-OHD: PG 28.3 ± 9.8; CG 22.3 ± 8.5	Significantly lower serum Ca and Mg
McMurry et al., USA (1992) [32]	190 (91 F) 26 PKU	PG 1.9–25.5 CG 3–16	Cohort study PG (1–5 years): 581 ± 121 PG (6–11 years) 1041 ± 188 PG (>11 years): 1629 ± 170 μmol/L	Mean serum Ca, Mg (atomic absorption spectrophotometry), and P (calorimetry) in mmol/L and 25-OHD (protein binding radio assay) in nmol/L	Ca: (1–5 years) PG 2.35 ± 0.04; CG 2.40 ± 0.02; (6–11 years) PG 2.32 ± 0.02; CG 2.30 ± 0.02; (>11 years) PG 2.35 ± 0.1; CG 2.32 ± 0.05 Mg: (1–5 years) PG 0.74 ± 0.04; CG 0.86 ± 0.04 (*p* < 0.001); (6–11 years) PG 0.74 ± 0.04; CG 0.91 ± 0.04 (*p* < 0.001); (>11 years) PG 0.66 ± 0.04; CG 0.82 ± 0.04 (*p* <0.001) P: (1–5 years) PG 1.68 ± 0.13; CG 1.65 ± 0.06; (6–11 years) PG 1.45 ± 0.06; CG 1.13 ± 0.06(*p* < 0.02); (>11 years) PG 1.13 ± 0.06; CG 1.65 ± 0.06 25-OHD: (1–5 years) PG 49 ± 8; CG 83 ± 4 (*p* <0.01); (6–11 years) PG 81 ± 17; CG 66 ± 2; (>11 years) PG 63 ± 10; CG 67 ± 4	Significantly lower serum Mg and 25-OHD in 1–5 y patient group; significantly higher P in 6–11 y
Pérez-Dueñas et al., Spain (2002) [39]	97 28 PKU	PG 18 (10–33) CG 10–34	Cohort study	Mean serum Ca, Mg, and P (standard procedure) in mg/dL	Ca: PG 2.42 (2.22–2.69); CG 2.41 (2.22–2.65) Mg: PG 0.82 (0.69–0.97); CG 0.83 (0.64–0.98) P: PG 1.22(0.77–1.66); CG 1.29(0.82–1.95) (*p* = 0.006)	Significantly lower P
Nagasaka et al., Japan (2011) [40]	70 (43 F) 34 PKU	PG 20–35 CG 19–40	Cross-sectional	Mean serum PTH and 25-OHD (RIA) in pg/mL	PTH: FPG 37.5 ± 2.4; FCG 32.3 ± 3.5 (*p* < 0.05); MPG 36.5 ± 3.8; MCG 32.7 ± 3.7 25-OHD: FPG 18.7 ± 1.3; FCG 27.6 ± 2.1 (*p* < 0.001); MPG 22.2 ± 1.7; MCG 30.0 ± 2.6 (*p* < 0.01)	Significantly higher PTH and 25-OHD in FPG; significantly lower 25-OHD in MPG

Ca, calcium; CG, control group; CTX, F, female; FPG, female patient group; Mg, magnesium; MPG, male patient group; P, phosphorus; PG, patient group; PKU, phenylketonuria; PTH, parathyroid hormone; RIA, radioimmunoassay; y, years; 25-OHD: 25 hydroxyvitamin D. ^1^ Values represent the range, the mean (range) or the mean ± SD in years, as reported in the corresponding article. ^2^ Blood phenylalanine levels represent range as reported in the corresponding articles. ^3^ Values represent the mean ± SD or mean (range) as reported in the corresponding article.

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
