# Peer review of "Bone Status in Patients with Phenylketonuria: A Systematic Review"

_nutrients, 2020, doi:10.3390/nu12072154_

Round 1
Reviewer 1 Report
The Authors present a systematic review on a topic with great interest in the field of PKU. The paper is very well written, with an adequate methodology and with a clear message.
Despite this, some remarks can be raised:
- Introduction; lines 39-42: please add references on GMP and LNAA, or a global review paper that addresses these treatment approaches. Also, it would be important to refer also the Special Low protein Foods (and a reference) considering it is a crucial component of the Phe restricted diet for these patients.
- Page 2; line 47: the authors should state that micronutrient nutritional deficiencies will be highly dependent on the compliance / adherence to the treatment.
- page 3; 2.6 Data Extraction: I think it would be useful to state who were the first two authors selecting the papers and what was the percentage of conflicts observed.
- Page 4; lines 144-145: What was the reason for the inclusion of this additional article? It is a particular important article known by the authors? If yes, does it add a significant input to the results?
- Page 4; section 3.1 Study Characteristics: do authors think that it would be important to highlight that more recent studies can be measuring outcomes in patients with improved nutritional and clinical management practices? In the last decades it was verified a significant improvement of the quality and diversity of protein substitutes together with a more universal recognition of the need to keep patients under follow-up with better metabolic control. This main affect comorbidities risk in patients with PKU.
- Tables: Please add the information on the country where the study was done. This will help the reader to match the results with current known clinical practices in different countries. I think we can agree this will be an important determinant of bone health in patients with PKU, particularly when the disease effect is still not precisely identified.
- Tables: Please simplify the results column, in order to facilitate reading and taking out the main and most important results.
- Page 16, lines 316-317: based on this sentence, which is true, the authors should change the titles of the tables. In fact, there is no absolute knowledge to say that the PKU, the disease itself, will be the cause of abnormalities / negative outcomes.
As a final suggestion, I would underline, for the future perspectives, to add a sentence about the availability of protein substitutes and special low protein foods in different countries, since it will determine different levels of diet compliance and adherence which will influence micronutrient status and bone status.
Author Response
The Authors present a systematic review on a topic with great interest in the field of PKU. The paper is very well written, with an adequate methodology and with a clear message. Despite this, some remarks can be raised:
- Introduction; lines 39-42: please add references on GMP and LNAA, or a global review paper that addresses these treatment approaches. Also, it would be important to refer also the Special Low protein Foods (and a reference) considering it is a crucial component of the Phe restricted diet for these patients.
ANSWER: Thank you for your comment. We have included two references in relation to GMP, LNAA and Special low protein foods in the introduction.
- Page 2; line 47: the authors should state that micronutrient nutritional deficiencies will be highly dependent on the compliance / adherence to the treatment.
ANSWER: We have included the following sentence in the introduction: “and mineral bone disease [15,16 13,14]. Such micronutrient nutritional deficiencies will be highly dependent on the compliance to the treatment”.
- Page 3; 2.6 Data Extraction: I think it would be useful to state who were the first two authors selecting the papers and what was the percentage of conflicts observed.
ANSWER: According to your suggestions we have included the name of the two authors that selected the papers and the percentage of conflicts observed.
“Two authors independently (MJdC and CdL) collected data from the…… ….The remaining authors acted as arbitrators in cases in which there was a lack of consensus (less than 3% of conflicts observed)”.
- Page 4; lines 144-145: What was the reason for the inclusion of this additional article? It is a particular important article known by the authors? If yes, does it add a significant input to the results?
ANSWER: The additional article (A study of bone turnover markers in prepubertal children with phenylketonuria by Jadwiga Ambroszkiewicz) investigates the effect of low-phenylalanine diets on bone mineralisation status, comparing biochemical bone formation and resorption markers in prepubertal children with phenylketonuria with those of age-matched healthy controls. It was retrieved after manually reviewing the references of one of the articles included, despite it was indexed in Pubmed we didn’t find it in the original search.
- Page 4; section 3.1 Study Characteristics: do authors think that it would be important to highlight that more recent studies can be measuring outcomes in patients with improved nutritional and clinical management practices? In the last decades it was verified a significant improvement of the quality and diversity of protein substitutes together with a more universal recognition of the need to keep patients under follow-up with better metabolic control. This main affect comorbidities risk in patients with PKU.
ANSWER: We agree with your suggestion. We have highlighted the above-mentioned information by including the following sentences in the study characteristics paragraph:
“… It should be highlighted that in the last decade it was verified a significant improvement of the quality and diversity of protein substitutes together with a more universal recognition of the need to keep patients under follow-up with better metabolic control. This is important because more recent studies can be measuring outcomes in patients with improved nutritional and clinical management practices.”.
- Tables: Please add the information on the country where the study was done. This will help the reader to match the results with current known clinical practices in different countries. I think we can agree this will be an important determinant of bone health in patients with PKU, particularly when the disease effect is still not precisely identified.
ANSWER: Thank you for your comments that greatly contributes to improve the quality of the manuscript. We have the information of the country were the study was done.
- Tables: Please simplify the results column, in order to facilitate reading and taking out the main and most important results.
ANSWER: We have simplified the results column following your recommendations by grouping patients by age, deleting p values when there was not statistical significance and eliminating the third decimal in the deviations from the mean.
- Page 16, lines 316-317: based on this sentence, which is true, the authors should change the titles of the tables. In fact, there is no absolute knowledge to say that the PKU, the disease itself, will be the cause of abnormalities / negative outcomes.
ANSWER: According to your suggestions we have replaced the titles of the tables with the following ones:
Table 1. Bone mineral content in patients with phenylketonuria
Table 2. Bone formation in patients with phenylketonuria
Table 3. Bone resorption in patients with phenylketonuria
Table 4. Serum minerals and hormones in patients with phenylketonuria.
- As a final suggestion, I would underline, for the future perspectives, to add a sentence about the availability of protein substitutes and special low protein foods in different countries, since it will determine different levels of diet compliance and adherence which will influence micronutrient status and bone status.
ANSWER: We completely agree with your comments. Indeed, we have included the following sentence in the discussion:
“However, both PKU and a Phe-restricted diet may should lead to a more pronounced decrease in BMD in older patients and, therefore, to increased incidence of osteoporosis in adult PKU patients. It should be underlined that the availability of protein substitutes and special low protein foods in different countries will determine different levels of diet compliance and adherence, which will influence micronutrient status and bone status”.

Reviewer 2 Report
I find this paper to be relevant and publishable. Bone status in PKU has been a concern for quite some time. Just a few minor edits. For the last few years there are several products used for the treatment of PKU that contain Phe. Line 41 specifically states treatment with a "Phe-free amino acid mixture". This is not the case currently, would suggest using "low Phe medical foods or amino acid mixtures.
Line 66 omit "being".
line 103 "limiting" instead of limitation.
line 288, consider "patients over 8 years of age.
line 310- change Phe-free to low phe
line 319- Phe-restricted diet MAY lead to a more pronounced decrease...……….
Author Response
I find this paper to be relevant and publishable. Bone status in PKU has been a concern for quite some time. Just a few minor edits. For the last few years there are several products used for the treatment of PKU that contain Phe.
Line 41 specifically states treatment with a "Phe-free amino acid mixture". This is not the case currently, would suggest using "low Phe medical foods or amino acid mixtures.
ANSWER: Thank you for your comment. We already changed the sentence: low Phe amino-acid mixtures
Line 66 omit "being".
ANSWER: Thank you, we have omitted “being” in line 66.
Line 103 "limiting" instead of limitation.
ANSWER: We have replaced “limitation” with “limiting” in line 103.
Line 288, consider "patients over 8 years of age.
ANSWER: We have rewritten the sentence in line 28 following your suggestion.
Line 310- change Phe-free to low phe
ANSWER: We have changed Phe-free to low Phe amino acid mixture.
Line 319- Phe-restricted diet MAY lead to a more pronounced decrease...……….
ANSWER: We have rewritten the sentence then should switch to may.
